# Role of the Microbiome and Its Metabolites in Primary Sjögren’s Syndrome

**DOI:** 10.3390/microorganisms13091979

**Published:** 2025-08-25

**Authors:** Jazz Alan Corona-Angeles, Roxana Lizbeth Martínez-Pulido, Edith Oregon-Romero, Claudia Azucena Palafox-Sánchez

**Affiliations:** 1Doctorado en Ciencias Biomédicas (DCB), Centro Universitario de Ciencias de la Salud, Universidad de Guadalajara, Guadalajara 44340, Jalisco, Mexico; coronaangeles.jazzalan@gmail.com (J.A.C.-A.); roxanalizbeth.mtz.p@outlook.com (R.L.M.-P.); 2Instituto de Investigación en Ciencias Biomédicas (IICB), Centro Universitario de Ciencias de la Salud, Universidad de Guadalajara, Guadalajara 44340, Jalisco, Mexico; oregon_edith@hotmail.com

**Keywords:** primary Sjögren’s syndrome, microbiome, dysbiosis, immune response, short-chain fatty acids

## Abstract

Primary Sjögren’s syndrome (pSS) is a chronic, autoimmune rheumatic disease characterized by progressive lymphocytic infiltration of the exocrine glands, leading to inflammation and subsequent tissue damage. As a multifactorial disease, its etiology is complex, making it difficult to predict disease progression. Among the environmental factors implicated in pSS, the involvement of microorganisms has gained increasing attention. Since the launch of the Human Microbiome Project, growing evidence has highlighted the role of dysbiosis in the pathogenesis of various autoimmune diseases, including pSS. Shifts in the abundance of specific bacterial phyla can lead to corresponding changes in the levels of key microbial metabolites involved in tissue homeostasis and immune regulation—such as short-chain fatty acids (SCFAs), choline, taurine, serine, lactate, and tryptophan and their metabolites. Understanding the mechanisms by which these metabolites influence immune processes may provide deeper insights into the progression of the disease. Therefore, this review aims to explore the mechanisms through which microbiota-derived metabolites contribute to the pathophysiology of primary Sjögren’s syndrome.

## 1. Introduction

Primary Sjögren’s syndrome (pSS) is a systemic autoimmune disease predominantly affecting women, with a female-to-male ratio of approximately 9:1, and typically appears between the ages of 40 and 60. It is characterized by progressive lymphocytic infiltration of the exocrine glands, leading to chronic inflammation; glandular destruction; and, ultimately, systemic involvement affecting multiple organs and tissues [1]. The clinical spectrum encompasses glandular symptoms, such as xerostomia and xerophthalmia, as well as extra-glandular manifestations, including fatigue; musculoskeletal pain; fever; renal dysfunction; and, in some cases, the development of lymphomas [2].

Classification of pSS is based on the 2016 ACR/EULAR criteria, which include key immunological features such as the presence of anti-SSA/Ro and anti-SSB/La autoantibodies and focal lymphocytic sialadenitis in minor salivary gland biopsies [3]. However, these criteria do not comprehensively capture disease heterogeneity or progression. The etiology of pSS remains unclear and is thought to involve a complex interplay of genetic predisposition, immune dysregulation, and environmental factors.

Several environmental triggers have been associated with pSS, including viral infections (e.g., Epstein–Barr virus, cytomegalovirus, and coxsackievirus), adjuvant exposure (e.g., vaccines and silicone implants), altered vitamin D levels, and bacterial infections such as *Helicobacter pylori* and non-tuberculous mycobacteria [4].

Although bacterial infections have been previously implicated in the pathogenesis of pSS, it was not until the advent of the Human Microbiome Project (HMP) that the human microbiome itself emerged as a potential contributor to autoimmune disease. Recent findings suggest that microbial dysbiosis is present in pSS patients and may correlate with symptom severity and disease activity. However, the specific role of microbiome in the development and progression of pSS remains largely unresolved [5].

The microbiome is now recognized as a key producer of bioactive metabolites, including short-chain fatty acids (SCFAs), bile acids, tryptophan, choline metabolites, gases, and vitamins. These compounds are involved in essential physiological processes ranging from metabolic regulation and gene expression to immune modulation and cellular homeostasis [6].

Given these insights, this review aims to investigate the potential contributions of microbiota-derived metabolites from different anatomical niches to the pathogenesis of pSS, with particular emphasis on the molecular mechanisms through which they may exert pathogenic or regulatory effects.

This is a narrative review based on the guiding question, *What is the effect of microbiota-derived metabolites on the progression of primary Sjögren’s syndrome (pSS)?* It considers all relevant information available to date, using the keywords ‘primary Sjögren’s syndrome’, ‘dysbiosis’, ‘autoimmunity’, and ‘microbiota-derived metabolites’. The literature was retrieved from major databases such as PubMed and ResearchGate. The aim is to provide a comprehensive overview of the effects of these compounds on the disease, with the purpose of fostering reader interest and encouraging further research in this field.

## 2. Microbiome in Health: Oral, Ocular, Gut, and Blood

The role of the microbiome in health and disease has been increasingly emphasized by numerous studies. Depending on their anatomical location, microbiota can be classified into distinct types; among these, the oral, ocular, and intestinal microbiota appear to be primarily involved in the pathophysiology of pSS. Moreover, emerging evidence supports the existence of a blood microbiome, which may also contribute to systemic homeostasis [7].

In healthy individuals, microbial communities exist in a state of symbiosis with the host, playing a crucial role in maintaining homeostasis and regulating immune responses. However, disruption of this balance—known as dysbiosis—can impair physiological functions and promote the development of various diseases, including autoimmune disorders [7].

### 2.1. Oral Microbiome

First, it is essential to elucidate the composition of the oral microbiome as xerostomia is often the first clinically detectable manifestation of pSS. The oral microbiome comprises the microorganisms residing within the oral cavity, including the teeth, tongue, cheeks, gingival sulcus, tonsils, hard palate, and soft palate. In a healthy oral environment, 16S rRNA sequencing has shown that the predominant microbial genera include the following:Gram-positive cocci: *Abiotrophia*, *Peptostreptococcus*, *Streptococcus*, and *Stomatococcus*.Gram-positive bacilli: *Actinomyces*, *Bifidobacterium*, *Corynebacterium*, *Eubacterium*, *Lactobacillus*, *Propionibacterium*, *Pseudoramibacter*, and *Rothia*.Gram-negative cocci: *Moraxella*, *Neisseria*, and *Veillonella*.Gram-negative bacilli: *Campylobacter*, *Capnocytophaga*, *Desulfobacter Desulfovibrio*, *Eikenella*, *Fusobacterium*, *Haemophilus*, *Leptotrichia*, *Prevotella*, *Selenomonas*, *Simonsiella*, *Treponema*, and *Wolinella*.Fungi: *Candida*, *Cladosporium*, *Aureobasidium*, *Saccharomycetales*, *Aspergillus*, *Fusarium*, and *Cryptococcus*.Virus: Bacteriophages [8].

Among these, *Fusobacterium nucleatum* is reported as the most commonly detected microorganism [8].

### 2.2. Ocular Microbiome

In contrast, the ocular microbiome exhibits significantly lower complexity than the oral microbiome, primarily due to the eye’s reduced exposure to the external environment. In a healthy ocular surface, 16S rRNA analysis has identified bacterial genera such as *Staphylococcus*, *Corynebacterium*, *Streptococcus*, *Micrococcus*, *Kocuria*, and *Propionibacterium* (Gram-positive), as well as *Haemophilus* spp., *Neisseria* spp., *Pseudomonas* spp., *Acinetobacter*, *Sphingomonas*, and *Brevundimonas* (Gram-negative) [9].

Among these, the phylum *Actinobacteria* is the most predominant. Additionally, in contrast to the oral microbiome, no fungal or viral species have been identified on the ocular surface to date [9].

### 2.3. Gut Microbiome

The gut microbiome is the most extensive and, additionally, the most extensively studied among those mentioned. It was, in fact, the first to be explored by HMP in 2007. In a healthy intestine, up to 100 trillion bacteria can be present, modulating intestinal health through structural, biochemical, and immunological mechanisms. Among the predominant bacterial phyla, *Firmicutes* is the most abundant, followed by *Bacteroidetes* and *Actinobacteria* [10]. In contrast to other microbiomes, identifying bacterial genera consistently associated with a healthy population is more challenging due to the high interindividual and population-level diversity [11].

Nevertheless, genera such as *Lactobacillus* and *Bifidobacterium* are known to play a critical role in stimulating mucin production. Similarly, *Escherichia* and *Bacteroides* species are capable of synthesizing vitamin K, while anaerobic bacteria from the genus *Clostridium* produce important regulatory metabolites, SCFAs [11].

### 2.4. Blood Microbiome

Finally, it is important to highlight a newly proposed microbiome class that, until recently, was considered implausible: the blood microbiome. Conventionally, blood has been regarded as a sterile environment. However, emerging evidence of microbial or pathogenic genetic material in the bloodstream has led to the conceptualization of a blood-associated microbiome [12].

To date, *Staphylococcus* spp. has been identified as a commonly detected genus in blood samples. Other potential microbial signatures include the phylum *Proteobacteria* and *Cutibacterium acnes*. Nevertheless, similar to the gut microbiome, the blood microbiome exhibits high interindividual variability, making it difficult to define a standardized blood microbiota [12].

This was supported by the findings of Tan et al. [13], who performed DNA sequencing of samples from 9770 healthy individuals using the Illumina HiSeq X platform (Illumina, San Diego, CA, USA). They reported the presence of up to 117 microbial species in the blood, including 110 bacteria, 5 viruses, and 2 fungi, with replication signatures from the genera *Alcaligenes*, *Caulobacter*, *Bradyrhizobium*, and *Sphingomonas*. Nevertheless, although these findings provide valuable insights into the potential composition of the blood microbiome, they should be interpreted with caution. It remains unclear whether these microorganisms are intrinsic to the bloodstream or originate from other tissues. Therefore, further studies are required to fully elucidate the components of this emerging microbiome.

## 3. Dysbiosis in pSS

The primary site of investigation regarding dysbiosis in pSS is the oral microbiome, where numerous studies have reported significant compositional changes. In 2016, Li M. et al. analyzed the oral microbiota of patients using high-throughput sequencing techniques and observed a reduction in the phylum *Proteobacteria*, accompanied by an increase in the genera *Leucobacter*, *Delftia*, *Pseudochrobactrum*, *Ralstonia*, and *Mitsuaria* [14].

In addition, research teams led by de Paiva C.S. et al. [15] and Siddiqui H. et al. [16], who characterized the microbiota through 16S rRNA gene sequencing, reported an increase in the genus *Streptococcus*, along with a reduction in genera such as *Leptotrichia*, *Fusobacterium*, *Fretibacterium*, and *Porphyromonas*. These bacterial increases have been previously associated with dental caries and may be linked to glandular tissue damage.

Regarding the ocular surface, Kim Y.C. et al. [17] reported that the Shannon diversity index was significantly lower in the pSS group, indicating reduced microbial diversity. Furthermore, 16S rRNA bacterial gene analysis, performed via sequencing on the Illumina MiSeq platform, revealed a decreased abundance of *Actinobacteria* and *Corynebacterium* in pSS patients, suggesting that reduced ocular microbiota diversity may be involved in the pathophysiology of the disease.

These findings were later confirmed by Song H. et al. [18], who reported similar trends in conjunctival swab samples. High-throughput sequencing of the V3–V4 region of the 16S rRNA gene demonstrated a reduction in the microbial diversity of the conjunctival sac, with a marked decrease in the genus *Bacillus* compared with other detected genera, including *Acinetobacter*, *Staphylococcus*, *Corynebacterium*, and *Clostridium*. These results further support a close association between ocular dysbiosis and the pathogenesis of Sjögren’s syndrome.

In the gut microbiome, the *Firmicutes*/*Bacteroidetes* (F/B) ratio is frequently used as an indicator of microbial balance. Recent studies have shown a decreased F/B ratio—indicating suppressed *Firmicutes* and the expansion of conditionally pathogenic *Bacteroidetes*—suggesting dysbiosis that may contribute to the pathogenesis of pSS [19]. This was further supported by Mandl T. et al. [20], who, through 16S rRNA sequencing of 24 stool samples from patients with pSS, observed a significantly reduced microbial diversity compared with healthy subjects (HS). Specifically, marked reductions in the genera *Alistipes* and *Bifidobacterium* were identified, which correlated with higher disease activity (assessed via the Sjögren’s Syndrome Disease Activity Index), lower complement levels, and elevated fecal calprotectin, a proposed biomarker for intestinal inflammation. Notably, reduced *Alistipes* has also been reported in psoriatic arthritis and Crohn’s disease, while *Bifidobacterium* levels were lower in both rheumatoid arthritis (RA) and Crohn’s disease, supporting their potential protective roles in autoimmune disorders [20].

Moreover, several studies have reported a significant reduction in the genus *Faecalibacterium*, particularly *Faecalibacterium prausnitzii*, one of the main butyrate-producing bacteria in the gut. Concurrently, an increased abundance of potential pathogens, such as *Escherichia/Shigella*, *Enterobacter*, and *Streptococcus*, has been observed, all of which are associated with moderate to severe infections and may further impair the quality of life in patients. Also, reduced levels of other butyrate-producing genera including *Bacteroides fragilis*, *Lachnoclostridium*, *Roseburia*, *Lachnospira*, and *Ruminococcus* have been identified, highlighting the functional significance of this SCFA in the pathophysiology of pSS [21].

With respect to the blood microbiome, no specific alterations have been reported in pSS to date. However, analyses in other autoimmune diseases, such as systemic lupus erythematosus (SLE) and RA, have revealed enriched levels of genera such as *Desulfoconvexum*, *Desulfofrigus*, *Desulfovibrio*, *Draconibacterium*, *Planococcus*, *Psychrilyobacter*, and the phylum *Gemmatimonadetes* in SLE. *Planococcus* has been associated with elevated plasma autoantibody levels, and exposure of heat-killed *Planococcus* to peripheral blood mononuclear cells (PBMCs) from SLE patients resulted in increased production of TNF-α, IL-1β, and IL-6, indicating a proinflammatory role in SLE pathogenesis [22].

In RA, Hammad D.B.M. et al. identified through sequencing four phyla—*Proteobacteria*, *Firmicutes*, *Bacteroidetes*, and *Actinobacteria*—as being more abundant in RA patients than in HS [23]. Furthermore, the genera *Halomonas* and *Shewanella* were found to be significantly increased in RA patients compared with control subjects.

Specifically, *Halomonas* has drawn attention due to its association with inflammatory markers, such as IL-1β, in the salivary microbiome—an anatomical site of high relevance in the pathophysiology of pSS. IL-1β is a key proinflammatory cytokine that is also implicated in RA, suggesting a potential role of this genus in autoimmune processes [24].

Despite these observations, there is currently no direct evidence of blood microbiome alterations in pSS. However, the findings from RA may offer useful insights into possible microbial shifts in pSS and guide future investigations into how bacterial metabolites may influence disease development. A summary of microbiome alterations reported in pSS and other autoimmune diseases is provided in Table 1.

## 4. Bacterial Metabolites and Their Importance in pSS

Based on the aforementioned evidence, it is clear that multiple bacterial genera may play a significant role in the development and progression of pSS. However, the major challenge lies in understanding how the upregulation or downregulation of these microbial populations influences the pathophysiology of the disease [30].

This question can be partially addressed by investigating the metabolites produced by these specific bacterial phyla and genera, many of which have been shown to modulate key immunological, metabolic, and tissue-related functions. Therefore, exploring these metabolic pathways offers a foundation to propose potential mechanisms through which microbial imbalance may directly or indirectly contribute to the onset and progression of pSS.

### 4.1. Short-Chain Fatty Acids (SCFAs)

First, the downregulation of bacterial genera such as *Faecalibacterium prausnitzii*, *Bifidobacterium*, *Bacteroides*, *Fusobacterium*, and *Bacillus* highlights the essential role of SCFAs in pSS. These bacteria are known for their ability to produce or stimulate the release of SCFAs, particularly butyrate, which has been most strongly associated with the pathophysiology of pSS. Butyrate is a saturated monocarboxylic acid derived from dietary fiber fermentation and is capable of modulating immune responses by inducing regulatory T cells (Tregs) [31].

The first insights into the protective role of butyrate in pSS came from the study by Kim D.S. et al., who used the NOD/ShiLtJ mouse model (a well-established animal model for pSS) [32]. Mice were administered varying concentrations of butyrate intraperitoneally over a 10-week period. The treatment promoted B10 cell expansion and reduced IL-17-producing T-cell populations. This effect was attributed to butyrate-mediated histone deacetylase (HDAC) inhibition and downregulation of *NR1D1*, a circadian rhythm-associated gene involved in Th17 cell differentiation. Butyrate also reduced Th17 cell infiltration and disease progression.

These findings were further supported by Yang L. et al., who examined the relationship between the gut microbiome and fecal metabolome in pSS patients using 16S rRNA sequencing and LC–MS-based metabolomics [33]. The authors observed a significant decrease in *Faecalibacterium* abundance, which correlated with reduced production of SCFAs such as butyrate, disrupting the balance between pro- and anti-inflammatory factors and promoting inflammation. Butyrate plays a key role in enhancing intestinal mucosal immunity by stimulating Treg cell function and IL-10 secretion, while simultaneously inhibiting proinflammatory cytokines such as IL-2, IL-8, and TNF.

Propionate has also emerged as a key SCFA in the context of pSS. Woo J.S. et al. reported that daily administration of *Lactobacillus acidophilus* (50 mg/kg) for 12 weeks in female NOD/ShiLtJ mice alleviated pSS-like symptoms in these animals [34]. Furthermore, the abundance of SCFA-producing bacteria, particularly propionate-producing species, increased following supplementation. Propionate production by *L. acidophilus* reduced lymphocytic infiltration in the salivary glands and altered the Th17/Treg ratio by decreasing Th17 cells and increasing Treg cells in the spleen. This effect was mediated by increased expression of STIM1, a negative regulator of STING, thereby reducing STING activity and type I interferon production—both crucial factors in pSS pathogenesis.

While most studies have focused on gut-derived or animal models, it is also critical to assess SCFAs at the level of the salivary glands. In this regard, Alt-Holland A. et al. employed nuclear magnetic resonance (NMR) spectroscopy to determine the composition of saliva samples from pSS patients and HS, identifying significant differences in salivary metabolomic profiles between the two groups [35]. Notably, butyrate and propionate levels were significantly reduced in pSS patients, and although acetate levels did not reach statistical significance, its potential biological relevance cannot be ruled out due to the shared receptor-binding mechanisms among SCFAs.

In summary, the evidence indicates that SCFAs exert anti-inflammatory effects by reducing lymphocytic infiltration in the glands. This effect is mediated through their specific receptors, GPR41 and GPR43, which modulate the expression of proteins involved in the differentiation of Th17 and Treg lymphocytes (Figure 1a) [35,36].

### 4.2. Choline, Trimethylamine (TMA), and Trimethylamine-N-Oxide (TMAO)

Choline is a saturated quaternary amine that can be derived from the catabolism of the phospholipid phosphatidylcholine. In this context, elevated choline levels are considered indicative of cellular membrane damage, reflecting the breakdown of membrane phospholipids. Wei Y. and Asbell P.A. identified that secretory phospholipase A2 group IIa (sPLA2-IIa) is involved in ocular surface inflammation in patients with dry eye disease. Furthermore, they demonstrated that sPLA2-IIa–mediated inflammation occurs through cooperative interactions with the proinflammatory cytokines TNF-α and IL-1β. The connection between sPLA2-IIa and choline lies in the enzymatic degradation of phosphatidylcholine to phosphocholine, which is subsequently converted into choline by choline kinase [37].

In addition to choline metabolism associated with cellular damage, various bacteria within the local microenvironment—particularly the gut microbiota—can hydrolyze membrane phosphatidylcholine via phospholipase D, further enhancing the release of choline. This choline is subsequently converted into trimethylamine (TMA), which is transported to the liver and oxidized to trimethylamine N-oxide (TMAO) by flavin-containing monooxygenase 3 (FMO3). Notably, certain intestinal bacteria are also capable of directly synthesizing TMAO [38,39].

Elevated TMAO levels have been linked to chronic inflammation and implicated in the pathogenesis of several autoimmune diseases, including SLE, systemic sclerosis, and psoriatic arthritis [40,41]. In the context of pSS, TMAO is proposed to contribute to tissue damage mechanisms by inducing oxidative stress and promoting the generation of reactive oxygen species (ROS). These ROS can damage DNA, proteins, and lipids, facilitating cellular injury and the exposure of autoantigens—key events in the perpetuation of autoimmunity. Additionally, TMAO has been shown to upregulate proinflammatory cytokines such as IL-1β, IL-6, and TNF-α, thereby exacerbating the inflammatory response characteristic of pSS (Figure 1b) [42].

### 4.3. Taurine

Taurine is an amino acid whose primary physiological function is to serve as a precursor in the synthesis of bile salts. However, several studies have shown that it also exerts a cytoprotective effect on ocular tissues. All retinal cells—from the outer and inner nuclear layers to the ganglion cell layer, and apparently also radial glial cells (Müller cells)—absorb taurine from the extracellular environment. Taurine depletion through inhibition of the taurine transporter (TauT) leads to ganglion cell loss along with degenerative changes in the retina [43].

This effect is attributed to secondary taurine metabolism, which produces haloamines such as taurine chloramine (TauCl) and taurine bromamine (TauBr). These compounds inhibit oxidative stress by interacting with the NADPH complex and simultaneously upregulate transcription factors involved in intracellular antioxidant mechanisms, such as Nrf2 (Figure 1c) [44]. These findings highlight the essential role of taurine in maintaining ocular tissue integrity. Nevertheless, the evidence is somewhat contradictory as elevated levels of taurine have been detected in the tears of patients with pSS. This increase may reflect a compensatory mechanism, wherein taurine metabolism is upregulated to mitigate tissue damage, with ocular cells themselves acting as the primary source of taurine rather than the local microbiome [36,44].

The previously mentioned findings have also been observed in the salivary glands, where taurine plays a crucial role in the cellular response to osmotic stress by regulating volume changes and the final composition of saliva through sodium flux. As noted earlier, patients with pSS exhibit significantly higher salivary taurine levels compared with HS. This elevated concentration may indicate oxidative stress in the glandular tissues, potentially contributing to the progression of tissue damage [45].

In any case, evaluating the origin of taurine and its role in pSS may offer a promising direction for future research.

### 4.4. Serine (Another Important Amino Acid)

Metabolomic analyses of tear fluid from patients with pSS have reported reduced levels of this amino acid. Serine plays a critical physiological role as a precursor in the synthesis of glutathione, one of the body’s most essential endogenous antioxidants. Several studies have demonstrated that antioxidant mechanisms decline with age, while oxidative stress progressively increases [46].

This increase in oxidative stress may contribute to disease progression as the Ro52/SSA protein has been identified as a peroxide-sensitive signaling molecule. Ro52/SSA can translocate from the cytoplasm to the nucleus via the mitogen-activated protein kinase (MAPK) pathway. In this context, it is plausible to hypothesize that serine depletion—and the resulting reduction in glutathione synthesis—may lead to excessive activation of MAPK pathways, ultimately resulting in the overexpression of Ro52/SSA, followed by cellular cytolysis. This process may promote the exposure of intracellular components to the immune system, facilitating immune sensitization and the subsequent production of autoantibodies (Figure 1d) [47].

Furthermore, the role of oxidative stress in disease progression has been reinforced by studies such as that by Ryo, K. et al., who reported elevated levels of 8-hydroxy-2′-deoxyguanosine (8-OHdG), a marker of oxidative DNA damage, in the saliva of patients with pSS compared with HS [48]. These findings underscore the contribution of oxidative stress and the loss of antioxidant precursors to the advancement of tissue damage in this condition.

### 4.5. Lactate

Lactate is a byproduct of glucose fermentation that has been found to be elevated in patients with pSS. The role of lactate in disease progression was evaluated by Xu J. et al., who investigated its effects in A253 cell lines (a human submandibular gland tumor cell line) and in NOD/Ltj mouse models [49]. Their findings demonstrated that lactate induces mitochondrial DNA (mtDNA) damage, leading to its leakage and subsequent activation of the cyclic GMP-AMP synthase-stimulator of interferon genes (cGAS–STING) pathway. This activation triggers the NF-κB signaling cascade and promotes the production of proinflammatory cytokines such as IL-6 and IL-8, along with the activation of the interferon regulatory pathway, resulting in the release of IFN-α, IFN-β, and TNF-α. These results offer compelling insight into the role of lactate metabolism in the pathogenesis of pSS, particularly considering that interferon signaling is a hallmark of the disease’s etiology (Figure 1e).

### 4.6. Tryptophan and Kynurenine

Tryptophan, an essential amino acid, is metabolized by both host enzymes and intestinal microbiota into various bioactive compounds, including kynurenine. In patients with Sjögren’s syndrome, elevated tryptophan catabolism and increased kynurenine levels have been reported, which may contribute to dysregulated immune responses and exacerbate disease severity [50].

Pertovaara et al. reported that patients with pSS exhibited a significantly increased rate of tryptophan degradation compared with healthy blood donors, along with elevated serum concentrations of kynurenine [51]. These findings were corroborated by Eryavuz Onmaz D. et al., who also observed significantly increased kynurenine levels in patients with moderate disease activity relative to HS [52].

This phenomenon is likely linked to the chronic inflammatory state characteristic of autoimmunity, which promotes the activation of indoleamine 2,3-dioxygenase (IDO), thereby enhancing the production of metabolites along the kynurenine pathway. Interestingly, a bidirectional relationship has been described between IDO and the gut microbiota: while IDO may exert immunosuppressive effects in the gastrointestinal tract by regulating microbial metabolism and immune reactivity, an altered gut microbiota may, in turn, affect the kynurenine pathway and IDO activity by modulating tryptophan availability [53].

Increased IDO activity and kynurenine production may represent a compensatory immune mechanism aimed at regulating chronic antigenic stimulation, possibly reflecting an attempt to suppress the ongoing T-cell–mediated autoimmune response (Figure 1f) [54].

Although the dysregulation of tryptophan metabolism, increased kynurenine levels, and altered IDO activity are well documented in pSS, these changes may not be specific to the disease. Nonetheless, further investigation is warranted to determine whether these metabolic alterations serve as functional biomarkers of tryptophan catabolism–dependent T-cell–mediated immunoregulatory mechanisms in autoimmune disorders [54].

### 4.7. Additional Metabolites of Interest

Metabolomic analyses have revealed significant alterations in various other metabolites in patients with pSS. Notably, altered concentrations of 5-aminopentanoate and fucose have been detected in saliva; lysophosphatidylcholines C18:1 and C18:2 and sphingomyelins C16:0 and C16:1, as well as aspartate and dopamine, have been identified in the ocular surface and tear fluid; and dysregulated levels of aflatoxin M1, glycolic acid, L-histidine, and phenylglyoxylic acid have been found in the gut of affected individuals. These compounds may represent candidate biomarkers of tissue damage and could serve as starting points for further research aimed at elucidating their protective or pathogenic roles in pSS [33,34,36].

## 5. Clinical Applications

Based on the aforementioned evidence, multiple beneficial effects have been associated with microbiota-derived metabolites. Therefore, the direct use of these metabolites, or therapies aimed at restoring the microbiota capable of producing them, could have a positive impact on the management of pSS and other autoimmune diseases.

SCFAs are among the most promising metabolites for clinical application. In NOD/ShiLtJ mouse models, sodium propionate supplementation has been shown to significantly reduce lymphocytic infiltration, leading to slower progression of glandular damage and decreased levels of pro-inflammatory cytokines (IL-6, IL-17, and TNF), including type I interferon, within the glandular tissue [34].

These findings are further supported by multiple studies in animal models of other autoimmune diseases, where administration of SCFAs—such as sodium butyrate, methyl butyrate, sodium propionate, and acetate—has demonstrated immunoregulatory effects. These include the expansion of Treg cells, attenuation of tissue damage progression, significant reduction in autoantibody levels, and enhanced expression of anti-inflammatory cytokines such as IL-10 [55].

Further research is required to confirm the safety of SCFAs as supplements in the management of autoimmune diseases such as pSS. However, we consider that the combined use of probiotics and prebiotics, together with a low-carbohydrate, high-fiber diet supplemented with specific amino acids, SCFAs, polyunsaturated fatty acids such as omega-3, and vitamins D and E, could represent a promising therapeutic approach to improve symptoms and quality of life in individuals with pSS. This strategy may promote the restoration of beneficial microbiota, reduce the production of lactate and ROS, and modulate the immune response, particularly by rebalancing the Th17/Treg cell ratio and regulating the expression of pro-inflammatory cytokines [56,57].

## 6. Concluding Remarks

Microbiota-derived metabolites play a crucial role in immune regulation and tissue homeostasis. Therefore, understanding the mechanisms by which these metabolites are altered in pSS is essential as they may serve as valuable biomarkers of tissue damage and disease progression.

Moreover, the growing interest in characterizing the blood microbiome opens new avenues for comparative analyses between patients with pSS and HS at both microbiome and metabolome levels. Such studies may provide deeper insights into the systemic effects of microbial metabolites in autoimmunity.

Undoubtedly, microbiota-derived metabolites hold significant promise as biomarkers for monitoring the progression of autoimmune diseases, including pSS. However, additional research is necessary to determine the extent to which these compounds exert long-term effects on disease development and severity. From our perspective, elucidating the specific roles of these metabolites in pSS offers a valuable opportunity to identify novel molecular targets associated with disease risk, progression, and potential therapeutic interventions in this complex autoimmune condition.

## Figures and Tables

**Figure 1 microorganisms-13-01979-f001:**
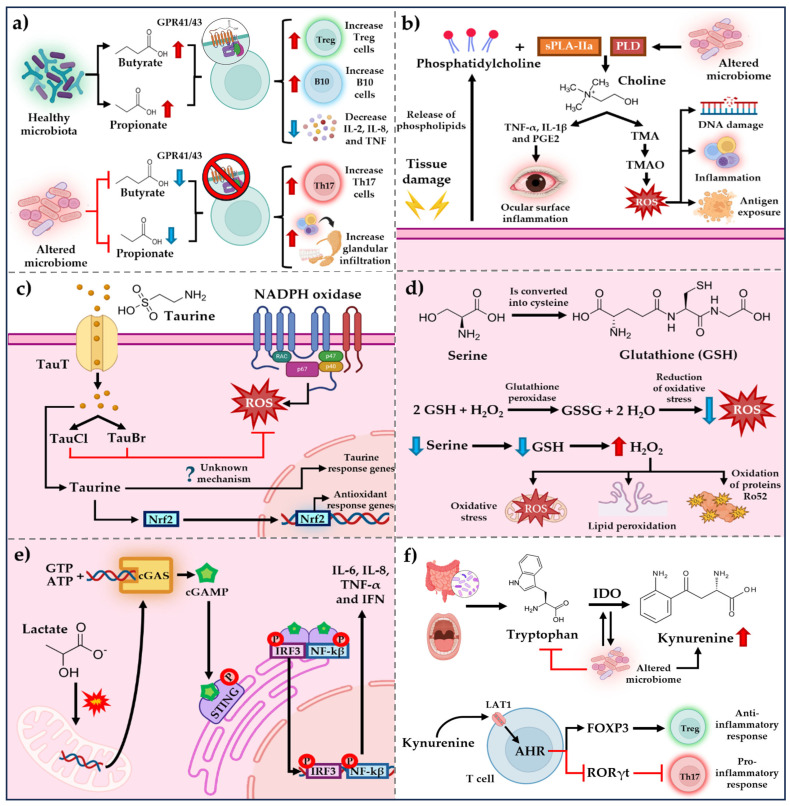
Summary of possible mechanisms of microbiota-derived metabolites in pSS. (**a**) SCAFs can regulate the immune response through their specific receptors GPR41 and GPR43. (**b**) Choline, TMO, and TMOA metabolites exert an effect on inflammation and oxidative stress. (**c**) Antioxidant mechanism of taurine and its metabolites. (**d**) Antioxidant mechanism of serine and the progression of damage after its depletion. (**e**) Mechanism of lactate in the antagonization of inflammatory responses mediated by cGAS. (**f**) Mechanism of tryptophan and kynurenine in the regulation of cellular immune response. PLD, Phospholipase D; sPLA2-IIa, secretory phospholipase A2 group IIa; TMA, Trimethylamine; TMAO, Trimethylamine-N-Oxide; TauT, Taurine transport; TauCl, Taurine chloramine; TauBr, Taurine Bromamine; ROS, Reactive Oxygen Species; Nrf2, Nuclear Factor Erythroid 2-elated Factor 2; GSH, Glutathione; GSSG, Glutathione Disulfide; cGAS, Cyclic GMP-AMP Synthase; cGAMP, Cyclic GMP-AMP; STING, Stimulator of Interferon Genes; IRF3, Interferon regulatory factor 3; NF-κB, Nuclear factor kappa-light-chain-enhancer of activated B cells; TNF-α, Tumor Necrosis Factor; IFN Interferon; LAT1, Large Neutral Amino Acid Transporter; IDO, Indoleamine 2,3-dioxygenase; AHR, Aryl Hydrocarbon Receptor; **↑**, increase; **↓**, decrease.

**Table 1 microorganisms-13-01979-t001:** Summary of the phyla and genera of the different microbiomes found in HS and patients with pSS and other autoimmune diseases.

	Oral Microbiome 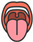	Ocular Microbiome 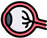	Gut Microbiome 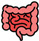	Blood Microbiome 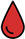
Phyla and genera in HS	*Abiotrophy*, *Peptostreptococcus*, *Streptococcus*, *Stomatococcus*, *Actinomyces*, *Bifidobacterium*, *Corynebacterium*, *Moraxella*, *Neisseria*, *Veillonella*, *Hemophilus*, *Leptotrichia*, *Prevotella*, *Selemonas*, *Treponema*, *Wolinella*, and *Fusobacterium nucleatum* [8].	*Staphylococcus*, *Corynebacterium*, *Streptococcus*, *Micrococcus*, *Kokuria*, *Propionibacterium*, *Haemophilus* spp., *Neisseria* spp., *Pseudomonas* spp., *Acinetobacter*, *Sphingomonas*, and *Brevundimona* [9].	*Firmicutes*, *Bacteroidetes*, *Actinomycetes*, *Faecalibacterium prausnitzii*, *Proteobacteria*, *Eubacterium*, *Ruminococcus*, *Lactobacillus*, *bifidobacterium*, *Escherichia*, *Bacteroides*, *Saccharomyces*, and *Clostridium* [11].	*Staphylococcus* spp. *Proteobacteria*, *Cutibacterium acnes*, *Alcaligenes*, *Caulobacter*, *Bradyrhizobium*, and *Sphingomonas* [12].
Phyla and genera in patients with pSS	**↑***Proteobacteria* and *Streptoccocus***↓** *Leucobacter*, *Delftia*, *Pseudochrobactrum*, *Ralstonia*, *Mitsuaria*, *Fusobacterium*, *Fretibacterium*, and *Porphyromonas* [14,15,16].	**↑***Acinetobacter*, *Corynebacterium*, and *Geobacillus***↓***Bacillus* spp. [17,18].	**↑***Escherichia/Shigella* and *Streptococcus***↓** *Alistipes*, *Bifidobacterium Faecalibacterium prausnitzii*, *Bacteroides fragilis*, *Lachnoclostridium*, *Roseburia*, *Lachnospira*, and *Ruminococcus* [19,20,21].	Not reported, possible participation of *Actinomyces* and *Halomonas* genera [22].
Phyla and genera in patients with SLE	**↑***Lactobacillaceae*, *Veillonellaceae*, and *Moraxellaceae***↓***Corynebacteriaceae*, *Micrococcaceae*, *Phyllobacteriaceae*, *Methylobacteriaceae*, *Sphingomonadaceae*, *Halomonadaceae*, *Pseudomonadaceae*, and *Xanthomonadaceae* [25].	**↑***Actinobacteria*, *Firmicutes*, *Bacteroidetes*, *Corynebacterium*, *Streptococcus*, and *Prevotella***↓***Pelomonas* and *Herbaspirillum* [26].	**↑***Rhodococcus*, *Eggerthella*, *Klebsiella*, *Prevotella*, *Eubacterium*, and *Flavonifractor* [27].	**↑***Desulfoconvexum*, *Desulfofrigus*, *Desulfovibrio*, *Draconibacterium*, *Planococcus*, *Psychrilyobacter*, and *Gemmatimonadete* [22].
Phyla and genera in patients with RA	**↑***Porphyromonas gingivalis*, *A. actinomycetemcomitans*, *Cryptobacterium curtum*, *P. intermedia/Tannerella forsythia Prevotella*, and *Leptotrichia* [28].	**↑***Corynebacterium*, *Streptococcus*, and *Prevotella***↓***Pelomonas* and *Herbaspirillum* [27].	**↑***Prevotella copri*, *Lactobacillus* spp., *Lactobacillus salivarius*, *Collinsella*, and *Akkermansia***↓** *Bacteroidetes*, *Bifidobacteria*, *Eubacterium rectale*, and *Haemophilus* spp. [29].	**↑***Proteobacteria*, *Firmicutes*, *Bacteroidetes*, *Actinobacteria Halomonas*, and *Shewanella* [23].

**↑**, increase; **↓**, decrease; HS, healthy subjects; SLE, systemic lupus erythematosus; pSS, primary Sjögren’s syndrome; RA, rheumatoid arthritis.

## Data Availability

No new data were created or analyzed in this study. Data sharing is not applicable to this article.

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
