# Peer review of "Role of the Microbiome and Its Metabolites in Primary Sjögren’s Syndrome"

_microorganisms, 2025, doi:10.3390/microorganisms13091979_

Round 1

Reviewer 1 Report

Comments and Suggestions for Authors

Review article : Role of the Microbiome and Its Metabolites in Primary Sjögren’s Syndrome

An interesting paper discusses the mechanisms through which metabolites derived from the microbiota contribute to the pathophysiology of primary Sjögren's syndrome.

Some revisions are needed.

  1. The abstract is correctly written.
  2. Keywords: Instead of SCFAs, write short-chain fatty acids.
  3. Section: Introduction
    It would be helpful to mention the method used for the literature search.
  4. The table and figures are clear and well-presented.
  5. A brief note on the clinical relevance of the above literature search results should be added.
  6. Section Conclusion: The sentences “Metabolomic analyses have revealed significant alterations in various other metabolites in patients with pSS...” (lines 426–434) should be placed elsewhere in the text, not in the conclusion.
  7. References: Each reference must be revised and formatted according to the journal’s guidelines.

Author Response

We would like to extend our sincere gratitude for the time and dedication devoted to reviewing our manuscript. The comments and suggestions provided were highly valuable in enhancing the quality and clarity of our work.

Following a thorough analysis of the observations raised, we have made the corresponding modifications, addressing each of the points outlined. We trust that the changes introduced adequately address the recommendations and contribute to strengthening the content of the manuscript.

Reviewer 1

Comments 1: The abstract is correctly written.

Response 1:  We appreciate the comment regarding the abstract. We believe that summarizing the most relevant points addressed in the article in a clear and concise manner is the most effective way to convey the overarching idea we aim to present in our review work. 

Comments 2: Keywords: Instead of SCFAs, write short-chain fatty acids.

Response 2:  We reviewed the observation provided and proceeded to replace the abbreviation SCFAs with “short-chain fatty acids.” The correction can be found in the keywords section, specifically between lines 26 and 27 of the manuscript. 

 Comments 3: Section: Introduction

It would be helpful to mention the method used for the literature search

Response 3:  We sincerely appreciate this comment and have addressed it as follows: between lines 63 and 69 in the Introduction section, we added a paragraph:

This is a narrative review based on the guiding question: What is the effect of microbiota-derived metabolites on the progression of primary Sjögren’s syndrome (pSS)? It considers all relevant information available to date, using the keywords primary Sjögren’s syndrome, dysbiosis, autoimmunity, and microbiota-derived metabolites. The literature was retrieved from major databases such as PubMed and ResearchGate. The aim is to provide a comprehensive overview of the effects of these compounds on the disease, with the purpose of fostering reader interest and encouraging further research in this field.

We included all studies meeting these criteria without applying temporal restrictions; however, we were rigorous regarding content, ensuring that it was coherent with the topic we intended to address. In addition, we prioritized research articles or reviews published in journals with an impact factor greater than 2, with the aim of maintaining the quality of the information presented in our work.

Comments 4: The table and figures are clear and well-presented.

Response 4: We appreciate the comments regarding the figure we prepared. We focused on integrating, within each section of the figure, the proposed mechanisms identified from the analysis of the articles collected for each microbiota-derived metabolite, with the aim of providing readers with a more visual representation of the potential mechanisms through which these metabolites are involved in the progression of pSS.

 Comments 5: A brief note on the clinical relevance of the above literature search results should be added.

Response 5: In compliance with the correction corresponding to this comment, and with the aim of further strengthening our work, we have added an additional section entitled 5. Clinical Applications, located between lines 446 and 470. This section briefly focuses on the use of certain microbiota-derived metabolites that have demonstrated immunoregulatory activity, such as short-chain fatty acids (SCFAs), proposed as supplementation in the management of the disease.

Furthermore, it highlights the need for additional research; however, we suggest that the combined use of these metabolites, or the enrichment of beneficial microbiota through prebiotics and probiotics, together with a diet rich in fiber and polyunsaturated fatty acids, may have a beneficial impact on patients’ quality of life.

Comments 6: Section Conclusion: The sentences “Metabolomic analyses have revealed significant alterations in various other metabolites in patients with pSS...” (lines 426–434) should be placed elsewhere in the text, not in the conclusion.

Response 6: We considered the observation to be pertinent and addressed it by adding a new section entitled 4.7. Additional Metabolites of Interest, in which we relocated the paragraph from the conclusion to this newly created subsection, positioned between lines 436 and 445. This section was adapted to highlight other potential microbiota-derived metabolites for which less information is currently available, but which may represent interesting candidates for analysis in future studies.

Comments 7: References: Each reference must be revised and formatted according to the journal’s guidelines.

Response 7: We appreciate this observation and have addressed it by adapting the reference formatting in accordance with the guidelines provided in the journal’s template. Specific changes were made to the references in Table 1; for instance, references [14], [15], and [16], which were previously listed separately, were merged as follows: [14–16]. These modifications were also applied to the references found in lines 269, 287, 290, 314, and 445.

Additionally, to ensure proper organization of the references, we used Mendeley software, and the format for each reference was revised to comply with the template’s specifications: Author 1, A.B.; Author 2, C.D. Title of the article. Abbreviated Journal Name Year, Volume, page range. In addition, the corresponding DOI was included for each reference to facilitate access and review by readers.

Sincerily, 

Claudia Azucena Palafox Sánchez

Reviewer 2 Report

Comments and Suggestions for Authors

This review presents a comprehensive synthesis of current findings regarding the role of microbiota-derived metabolites in Primary Sjögren’s Syndrome (pSS). It examines microbiome composition and dysbiosis across four anatomical niches (oral, ocular, gut, and blood) and explores how microbial metabolites—especially short-chain fatty acids (SCFAs), choline derivatives (TMAO), taurine, serine, lactate, and tryptophan-kynurenine products—may contribute to the immune dysregulation, tissue damage, and inflammation characteristic of pSS.

The article integrates insights from microbiology, immunology, metabolomics, and autoimmunity, making the topic highly relevant to the field of autoimmune microbiome research. It offers impressive coverage of microbial metabolites and their molecular pathways. However as the authors mentionned, further research is necessary to elucidate their protective or pathogenic roles in pSS. Additionally, more studies are required to determine the extent to which these compounds exert long-term effects on disease development and severity.

Table 1 is well-constructed and provides a good overview of pSS alongside other autoimmune pathologies such as SLE and RA.

Figure 1 effectively illustrates the possible mechanisms of microbiota-derived metabolites in pSS. The sections are clearly divided and logically ordered.

Overall, this is a valuable and thorough review covering an emerging area of immunometabolism in autoimmune disease. With structural refinement, inclusion of any missing visuals, and a sharper analytical focus, this manuscript could serve as a high-impact reference for microbiome researchers and clinicians working on pSS and related conditions.

Author Response

We would like to extend our sincere gratitude for the time and dedication devoted to reviewing our manuscript. The comments and suggestions provided were highly valuable in enhancing the quality and clarity of our work.

Following a thorough analysis of the observations raised, we have made the corresponding modifications, addressing each of the points outlined. We trust that the changes introduced adequately address the recommendations and contribute to strengthening the content of the manuscript.

Reviewer 2

 Comments1: This review presents a comprehensive synthesis of current findings regarding the role of microbiota-derived metabolites in Primary Sjögren’s Syndrome (pSS). It examines microbiome composition and dysbiosis across four anatomical niches (oral, ocular, gut, and blood) and explores how microbial metabolites—especially short-chain fatty acids (SCFAs), choline derivatives (TMAO), taurine, serine, lactate, and tryptophan-kynurenine products—may contribute to the immune dysregulation, tissue damage, and inflammation characteristic of pSS.

The article integrates insights from microbiology, immunology, metabolomics, and autoimmunity, making the topic highly relevant to the field of autoimmune microbiome research. It offers impressive coverage of microbial metabolites and their molecular pathways. However as the authors mentioned, further research is necessary to elucidate their protective or pathogenic roles in pSS. Additionally, more studies are required to determine the extent to which these compounds exert long-term effects on disease development and severity.

Table 1 is well-constructed and provides a good overview of pSS alongside other autoimmune pathologies such as SLE and RA.

Figure 1 effectively illustrates the possible mechanisms of microbiota-derived metabolites in pSS. The sections are clearly divided and logically ordered.

Overall, this is a valuable and thorough review covering an emerging area of immunometabolism in autoimmune disease. With structural refinement, inclusion of any missing visuals, and a sharper analytical focus, this manuscript could serve as a high-impact reference for microbiome researchers and clinicians working on pSS and related conditions.

Response 1: We sincerely thank the reviewer for their thoughtful and encouraging comments on our manuscript. We greatly appreciate the recognition of our efforts to integrate current findings from microbiology, immunology, metabolomics, and autoimmunity in the context of pSS.

We fully agree with the observation that further research is necessary to clarify the protective or pathogenic roles of microbiota-derived metabolites, as well as their potential long-term effects on disease development and severity. As noted in our discussion, the present review focused on those metabolites for which sufficient evidence is currently available; However, we anticipate that future studies will allow for the inclusion and deeper analysis of additional metabolites, enabling a more comprehensive review of their implications in pSS.

We are confident that the continued expansion of research in this area will contribute to a more complete understanding of the microbiome's role in the pathophysiology, progression, and management of pSS.

Sincerily,

Claudia Azucena Palafox Sánchez

Round 2

Reviewer 1 Report

Comments and Suggestions for Authors

 no comments